# iTRAQ-Based Quantitative Proteomics Analysis Reveals the Mechanism of Golden-Yellow Leaf Mutant in Hybrid Paper Mulberry

**DOI:** 10.3390/ijms23010127

**Published:** 2021-12-23

**Authors:** Fenfen Wang, Naizhi Chen, Shihua Shen

**Affiliations:** 1Key Laboratory of Plant Resources, Institute of Botany, The Chinese Academy of Sciences, Beijing 100093, China; wangfenfen@ibcas.ac.cn; 2University of Chinese Academy of Sciences, Beijing 100049, China

**Keywords:** chlorophyll, chloroplast, photosynthesis, ribosome, proteome

## Abstract

Plant growth and development relies on the conversion of light energy into chemical energy, which takes place in the leaves. Chlorophyll mutant variations are important for studying certain physiological processes, including chlorophyll metabolism, chloroplast biogenesis, and photosynthesis. To uncover the mechanisms of the golden-yellow phenotype of the hybrid paper mulberry plant, this study used physiological, cytological, and iTRAQ-based proteomic analyses to compare the green and golden-yellow leaves of hybrid paper mulberry. Physiological results showed that the mutants of hybrid paper mulberry showed golden-yellow leaves, reduced chlorophyll, and carotenoid content, and increased flavonoid content compared with wild-type plants. Cytological observations revealed defective chloroplasts in the mesophyll cells of the mutants. Results demonstrated that 4766 proteins were identified from the hybrid paper mulberry leaves, of which 168 proteins displayed differential accumulations between the green and mutant leaves. The differentially accumulated proteins were primarily involved in chlorophyll synthesis, carotenoid metabolism, and photosynthesis. In addition, differentially accumulated proteins are associated with ribosome pathways and could enable plants to adapt to environmental conditions by regulating the proteome to reduce the impact of chlorophyll reduction on growth and survival. Altogether, this study provides a better understanding of the formation mechanism of the golden-yellow leaf phenotype by combining proteomic approaches.

## 1. Introduction

Photosynthesis is one of the most important biological processes and harnesses solar energy to provide biochemical energy and oxygen for humans. Plant leaves are the primary organs of photosynthesis and have many chloroplasts in each mesophyll cell [1]. Chloroplasts are heavily involved in both the harvesting of light and the transduction of energy, making them important organelles in the photosynthetic cells of higher plants by supplying both carbon and energy. The chloroplasts consist of a complex thylakoid membrane system that embeds pigment-protein supercomplexes, including photosystem I (PSI), photosystem II (PSII), cytochrome b6f, and ATP-synthase [2]. The thylakoid membrane is in the space where PSI and PSII work and is responsible for harnessing light energy. The chlorophyll and carotenoid pigments in the light-harvesting antenna complex (LHC) capture most of the light energy [3]. LHC obtains the light energy needed to produce important photochemical reactions needed for photosynthesis and photoprotection once the available light energy is greater than the photosynthetic capacity [4]. Chlorophyll is distributed in chloroplast thylakoid membranes, is a major component of the light-harvesting complex, and plays an indispensable role in photosynthesis [5,6]. Leaf color is directly related to the development of chloroplasts and the production of photosynthetic pigments [7]. Therefore, leaf color mutants are widely used to identify regulatory mechanisms for chlorophyll biosynthesis, chloroplast development, and photosynthesis. Leaf color is an important plant trait and has attracted wide attention from researchers and the general public. Many reports have shown that most leaf color mutations are related to changes in chloroplast structure and function [8], chlorophyll biosynthesis and degradation mechanisms [9], and photosynthesis [10,11]. Different mutations have resulted in different leaf colors, like albino, golden-yellow, light green, yellow-green, and dark green. In plants, leaf color mutants have been successively found in recent years. For example, in rice, mutations in the genes that encode glutamyl-tRNA [12], mg-chelatase [13], and magnesium-protoporphyrin IX monomethyl ester cyclase [14] lead to changes in leaf color.

A hybrid paper mulberry (*Broussonetia kazinoki* × *Broussonetia papyrifera*), which belongs to the Moraceae family, has a rapid growth rate, high protein content, great adaptability to climates, and multi-resistance to pests and diseases [15]. Therefore, it has been widely used in papermaking, ecological restoration, and medicine [16]. The leaf of hybrid paper mulberry has a high total flavonoid and phenol content [17]. Hybrid paper mulberry plants have shown antioxidant and antifungal activities and have been used as a sort of forage, which is superior to compound feed. It has also helped reduce antibiotics and improve milk quality in China [18].

Transcriptomic analysis provides a holistic picture of networks of gene expression, while proteomic analysis relates these networks to protein products and provides additional information about posttranscriptional changes. Proteomics can be used for widespread, systemic analyses of the molecular regulatory mechanisms and protein networks associated with environmental responses and development in plants [19]. Proteomics has been used to analyze the coloring mechanism of leaves in certain higher plants. For example, proteomic analysis of leaf mutants and wild types identified a mechanism involved in changing the leaf color of Ginkgo mutants by Liu et al. [20]. Dong et al. found that the levels of protein expression affect the mechanisms causing chlorophyll deficiencies in the leaves of tea plants [21]. At present, one of the best mass spectrometry methods is isobaric tags for relative and absolute quantitation (iTRAQ), which measure the intensity of the peaks of reporter ions during MS/MS to produce a quantitative analysis [22]. Therefore, the high-throughput proteomic analysis of golden-yellow mutants in hybrid paper mulberry will not only uncover a potential mechanism for leaf color variation but contribute to a systematic analysis of the protein network of hybrid paper mulberry.

In this study, we compared the chloroplast structure, pigment content, photosynthetic properties, chlorophyll, and carotenoid synthesis precursors of golden-yellow mutant leaves and normal green leaves. We also investigated the effects of the proteome by iTRAQ technology combined with liquid chromatography and tandem mass spectrometry (LC-MS/MS) and identified the differentially accumulated proteins related to pigment biosynthesis and metabolism. Additionally, we measured the levels of five intermediate products related to the biosynthesis of chlorophyll and six compounds related to the biosynthesis of carotenoid to explore the molecular mechanisms associated with the biosynthesis of pigments in golden-yellow mutants. Additionally, we used quantitative real-time polymerase chain reaction (qRT-PCR) to confirm the expression of genes related to leaf coloration. This analysis of the differentially accumulated proteins and differential expression of genes between mutant and normal leaves could provide novel insights into the potential molecular mechanism underlying the mutant phenotype and provide a useful reference for leaf coloration studies in other plant species.

## 2. Results

### 2.1. Physiological Changes in Mutant Leaves

The mutant was a spontaneous mutant with a golden-yellow leaf phenotype and was smaller than the green leaf plant throughout the developmental stage (Figure 1A). To analyze the mechanism of leaf color formation, we determined the contents of chlorophyll (Chl), carotenoid (Car), and flavonoid in mutant and green leaves. The green leaf Chl a content was 2.33 mg g^−1^, which was higher than the mutant leaves (0.41 mg g^−1^). Similarly, the Chl b content of the mutant leaves was 0.03 mg g^−1^, which was lower than that of the green leaves (0.71 mg g^−1^). Decreases of chlorophyll b in mutant leaves exceeded that of chlorophyll a, resulting in a higher ratio of Chl a/b (Figure 1B). The Car content of mutant leaves decreased to 53.5% of that in the green leaves. However, the ratio of Car to Chl in the mutant leaves exceeded that in green leaves (Figure 1C). Additionally, the total flavonoid content of mutant leaves was 69 mg g^−1^ and was higher than that of green leaves (51 mg g^−1^) (Figure 2A). These results demonstrate that decreases in Car and Chl contents, increases in flavonoid contents, and alterations of the Car/Chl and Chl a/b ratios could be responsible for the golden-yellow color of mutant leaves.

In addition, we assessed the levels of Chl biosynthesis precursors in the mutant and green leaves to investigate the biochemical steps that were disrupted and compared seven precursors associated with the Chl biosynthesis metabolic process (Figure 2B). Further analysis demonstrated that the chlorophyll synthesis precursor 5-aminolevulinic acid (ALA), porphobilinogen (PBG), uroporphyrin III (Urogen III), and coproporphyrin III (Coprogen III) all significantly increased in mutant leaves. Additionally, the contents of protochlorophyllide (Pchlide), protoporphyrin (Mg-Proto), and protoporphyrin IX (proto IX) all significantly declined in the mutant leaves when compared to the green leaves. High ratios of Car/Chl led us to assess the vital carotenoid components in hybrid paper mulberry. The content of zeaxanthin in the mutant leaves was significantly higher than that in the green leaves, while the content of the other carotenoid components significantly decreased (Figure 2C).

### 2.2. Anatomical Features and Ultrastructure of Chloroplasts in Mutant Leaves

To estimate whether the difference in pigment contents is caused by changes in the chloroplast or leaf structure in mutants, we examined the leaf anatomy and the ultrastructure of the chloroplast of the green and mutant leaves. The thickness of the upper epidermis of mutant leaves decreased compared with the green leaves. Additionally, the palisade parenchyma intercellular spaces were greater in mutant leaves than in green leaves (Figure 3).

Chloroplasts are responsible for leaf greening, leading us to investigate the ultrastructure of the chloroplast in green and mutant leaves by TEM (transmission electron microscopy). The grana, which is comprised of thylakoids, is where chlorophyll and pigments related to photosystems are located. TEM analysis demonstrated that green leaves exhibited a typical chloroplast ultrastructure with distinct thylakoids stacking, starch granules, and osmiophilic granules. In contrast, the ultrastructure analysis of chloroplasts of mutant leaves showed ruptured thylakoid stacking, indistinct stromal lamella, and that some chloroplasts contained white vesicles. These results indicate that the anatomical structure and development of chloroplasts were impaired in the mutant leaves.

### 2.3. Photosynthetic Parameter Analyses in Mutant Leaves

Compared with green leaves, the analysis of gas-exchange measurements showed that the rate of net photosynthesis and stomatal conductance in mutant leaves decreased by approximately 37.1% and 17.5%, respectively. In contrast, intercellular CO_2_ concentration and transpiration rate increased by approximately 28.4% and 7.8%, respectively, in mutant leaves (Table 1).

The chlorophyll fluorescence parameters were measured to better understand observed decreases in the net photosynthesis rate in mutant leaves. The results demonstrated that levels of Fo, Fm, Fv, and Fv/Fm in mutant leaves were significantly lower than in green leaves. Compared with green leaves, Fo, Fv, Fm, and Fv/Fm levels in mutant leaves decreased by 10.27, 52.09, 62.52, and 27.50%, respectively (Table 2).

### 2.4. Quantitative Identification of Mutant Leaf Proteins Using iTRAQ

After combining the results of three independent biological replicates, the iTRAQ experiment produced 631,893 spectra (with the mutant and green leaves as materials). A total of 14,943 unique spectra and 15,804 matched spectra were identified by Mascot. We identified a total of 6157 unique peptides and 4766 proteins. The distribution of protein mass (Appendix A), number, and length of peptides (Appendix A) were provided. According to GO analysis, 1575, 853, and 2491 proteins were annotated as biological processes, cellular components, and functional molecules, respectively (Appendix A). The major biological function categories included photosynthesis, carbon metabolism processes, and nucleoside phosphate metabolic processes. A total of 3331 proteins between the green and mutant leaves were classified into 25 COG categories, of which the functional categories were translation, ribosomal structure, and biogenesis (13.0%); general function prediction (12.4%); post-translational modification, protein turnover, chaperones (10.7%); and carbohydrate transport and metabolism (9.6%) (Appendix A). To further investigate the biological functions of these proteins, 3016 were mapped to 19 pathways in the KEGG database. The pathways were metabolic pathways (70.3%), genetic information processing (23.1%), cellular processes (4.0%), environmental information processing (1.3%), and organismal systems (1.2%) (Appendix A).

### 2.5. Protein Differences between Green Leaves and Mutant Leaves

In this study, the abundance of 168 proteins were significantly changed (|log_2_FC| > 0.26; FC > 1.2 or FC < 0.83; [fold change, FC]; *p* < 0.05), with 51 upregulated and 117 downregulated between mutant and green leaves (Figure 4A,B). The protein expression patterns of both the green and mutant leaves were analyzed by hierarchically clustering the DAPs (Figure 4C). Of the identified proteins, 6, 21, and 90 proteins decreased by 0.50-, 0.67-, and 0.83-fold in the mutant, respectively. Additionally, the abundance of 0, 6, and 45 proteins increased by 2.0-, 1.5-, and 1.2-fold in the mutant, respectively (Appendix A). GO enrichment analysis was then used to group the differentially accumulated proteins into three categories: cellular components, biological process, and molecular function (Figure 5A). The representative major biological functional categories were cellular metabolic processes, photosynthesis, photosynthetic electron transport in photosystem II, dephosphorylation, and cell wall macromolecular catabolic processes. According to the different molecular functions, these proteins were primarily divided into tetrapyrrole binding, enzyme inhibitor activity, endopeptidase inhibitor activity, chlorophyll-binding, and chitin-binding. The COG database was used to classify the 58 differentially accumulated proteins into eight different groups (Figure 5B). The major functional categories were translation, ribosomal structure, and biogenesis (24.1%); carbohydrate, transport, and metabolism (22.4%); signal transduction mechanisms (15.5%); cell wall/membrane/envelope biogenesis (13.8%); coenzyme transport and metabolism (10.3%); defense mechanisms (6.9%); inorganic ion transport and metabolism (5.2%); and chromatin structure and dynamics (1.7%). The KEGG database was used to further analyze these differentially accumulated proteins, and we found that they were annotated in the metabolic pathways (35.8%), photosynthesis antenna proteins (10.6%), photosynthesis (10.6%), ribosome (4.9%), alanine, aspartate and glutamate metabolism (2.4%), endocytosis (2.4%), amino sugar and nucleotide sugar metabolism (2.4%), steroid biosynthesis (1.6%), porphyrin and chlorophyll metabolism (1.6%) and phenylalanine, tyrosine, and tryptophan biosynthesis (0.8%) (Figure 5C).

### 2.6. Identification of DAPs Related to Pigment Metabolism

The primary photosynthetic pigment is Chl, which is found in photosynthetic organisms and obtains light energy from the antenna systems by transferring electrons in the reaction centers of higher plants. In higher plants, chlorophyll synthesis requires a lot of enzymes, such as glutamyl tRNA reductase (GluTR), magnesium chelatase (Mgch), uroporphyrinogen decarboxylase (UROD), and coprogen III oxidase (CPO) [23]. Three subunits make up magnesium chelatase, ChlH, ChlI, and ChlD, which then form Mg-protoporphyrin IX from protoporphyrin IX. Studies in wheat mutants suggested that a block in Mg-chelatase activity was responsible for their mutant phenotype [24]. The Arabidopsis genomes uncoupled 4 (GUN4) gene, which encodes the ChlH subunit of Mg-chelatase, and account for the yellow phenotypes of this mutant [25]. UROD catalyzes the stepwise decarboxylation of the four acetate residues of Urogen III to form Coprogen III [23]. In this study, these three proteases associated with chlorophyll biosynthesis were changed. These proteins include uroporphyrinogen decarboxylase (UROD, Bp03g0811), tetrapyrrole-binding protein (GUN4, Bp13g0174), and magnesium chelatase H subunit (CHLH, Bp01g0004), and their abundance all increased in the mutant leaves in comparison with green leaves.

Carotenoids play an essential role in both photosynthesis and photoprotection. Because carotenoids can present a variety of colors, including yellow, orange, or red, they are one of the most important factors affecting the color of plant leaves [26]. In our dataset, one differentially accumulated protein was annotated as the key protein related to carotenoid biosynthesis. We found that zeaxanthin epoxidase (ZEP, Bp01g2472) was downregulated in mutant leaves compared with green leaves, and it may play an important role in golden-yellow leaf formation in hybrid paper mulberry. These results indicate that there were notable differences in the metabolism of green and mutant leaves.

### 2.7. Differentially Accumulated Proteins Involved in Photosynthesis and Photosynthesis-Antenna Biosynthesis

In the photosynthesis pathway, a total of 26 DAPs encoding core proteins of Photosystem I (PS I), Photosystem II (PS II), and the light-harvesting chlorophyll protein complex (LHC) were all downregulated in the mutant leaves (Figure 6). In PSI, the down-regulated DAPs included Photosystem I P700 chlorophyll an apoprotein A1 (PsaA), Photosystem I P700 chlorophyll an apoprotein A2 (PsaB), Photosystem I reaction center subunit II (PsaD), Photosystem I reaction center subunit III (PsaF), Photosystem I reaction center subunit V (PsaG), Photosystem I reaction center subunit VI (PsaH), Photosystem I reaction center subunit XI (PsaL), and Photosystem I reaction center subunit PsaK. In PSII, the down-regulated DAPs included Photosystem II protein D1 (PsbA), Photosystem II CP47 reaction center protein (PsbB), Photosystem II CP43 reaction center protein (PsbC), Photosystem II protein D2 (PsbD), and Photosystem II 10 kDa polypeptide (PsbR). In addition, 13 proteins encoding the chlorophyll a-b binding protein (LHCA1, LHCA2, LHCA3, LHCA4, LHCA5, LHCB1.1, LHCB1.2, LHCB2, LHCB3, LHCB4.1, LHCB4.2, LHCB5, LHCB6) were suppressed in mutant leaves. This indicates that suppressing photosynthetic proteins negatively affected the development of chloroplasts in mutant leaves.

### 2.8. Differentially Accumulated Proteins Involved in Ribosome Pathway

Up-regulated proteins were primarily related to oxidoreductase activity and binding categories, particularly DNA binding, RNA binding, and lipid binding (Figure 7). Heat shock proteins (HSPs) and ribosomal proteins are the two major components of these binding proteins. Compared with green leaves, some ribosomal proteins in mutant leaves were up-regulated, including four large chloroplastic ribosomal subunits (L15, L19B, L24, and L31A) and two small chloroplastic ribosomal subunits (S26A and S12B). HSPs, which include HSP70, form another protein family with up-regulated abundance in mutant leaves. In addition, proteins with up-regulated abundance include DNA replication protein, translation regulation factor, and DNA repair protein in mutant leaves.

### 2.9. QRT-PCR Analysis of the Differentially Accumulated Proteins

The qRT-PCR analysis was conducted to better understand the relationship between mRNA transcription and protein expression and to confirm the authenticity of the iTRAQ analysis. A total of 10 genes were determined to be involved in chlorophyll biosynthesis, photosynthesis, and photosynthesis-antenna biosynthesis (Figure 8). Nine genes had expression trends that were the same as their corresponding proteins, including PsaA, PsaB, PsbA, PsbB, UROD, ZEP, CCR, LHCA2, and LHCB4.2. However, only one gene (CHLH) showed the opposite trend from their corresponding protein expression level. The discrepancy between the transcription level and the abundance of the corresponding protein likely resulted from various post-translational modifications, including protein phosphorylation and glycosylation.

## 3. Discussion

### 3.1. DAPs Involved in Chlorophyll Synthesis and Chloroplast Development

Leaf color is a key plant feature, and its formation is strongly associated with the synthesis and transportation of chlorophyll. In higher plants, the color of a leaf depends on chlorophyll biosynthesis and chloroplast development. Thus, leaf color mutations are customarily chlorophyll-deficient mutations [27]. Chlorophyll includes chlorophyll a and chlorophyll b, which is mainly responsible for the capture and photosynthetic transformation of light and is distributed in the chloroplasts of the leaves. Chlorophyll is the most important pigment that can process light morphogenesis and organic accumulation in plants [28]. In this study, we found the chlorophyll-deficient mutant of hybrid paper mulberry. The mutant presents a golden-yellow leaf phenotype and is a spontaneous leaf color mutant. The content of chlorophyll a and b in mutant hybrid paper mulberry leaves is much lower than the content found in typical green leaves. This indicates that chlorophyll deficiencies can change leaf color at a physiological level. We hypothesize that the golden-yellow phenotype of mutant leaves is due to a lack of Chl, which is similar to the studies on *Lagerstroemia indica* [29], and *Ginkgo biloba* [30].

Chlorophyll, carotenoids, and flavonoids are the major components in leaves. We performed a comprehensive biochemical analysis and proteome profiling of mutant leaves and green leaves to better understand the mechanism of leaf color variations during complex biological processes at the proteomics level. Chl metabolism is processed by complex biological processes. Blocking any step in this process will lead to a decrease in Chl content, which changes the leaf color. Any mutation in Chl biosynthetic genes can affect Chl accumulation [31]. In our study, some Chl biosynthetic enzymes in mutant leaves have changed at both the mRNA and protein levels. Mgch, which is made up of the subunits ChlH, ChlD, and ChlI, was altered. Mgch is responsible for catalyzing the magnesium-insertion step during Chl synthesis. One study found that chlorophyll synthesis decreased when genes encoding Mg-chelatase subunits ChlD and ChlI were mutated, including the *rice* mutants *Chlorina-1* and *Chlorina-9* [8]. Therefore, we hypothesized that changes in Mgch could be due to Chl decreases in mutant leaves.

Photosynthesis occurs in the chloroplast. The thylakoid membranes are stacked into grana and regularly arranged in the chloroplasts. The presence of grana stacks suggests that compact light-harvesting machinery is effective at absorbing and converting light energy [32]. Most leaf color mutants demonstrate changes in their thylakoid membrane, as reported in previous studies. [33]. For instance, the results have shown that chloroplasts have a typical structure in *Ginkgo biloba* leaves, with small starch granules and obvious thylakoid membrane and stromal lamellae. However, the ultrastructural analysis of golden leaf mutant chloroplasts shows that the stromal lamellae are not clear, the thylakoid membrane is broken, and some chloroplasts contain irregular vesicles [30]. In this study, we found that the chloroplast structure in mutants was significantly different from normal green leaves, the structure of chloroplasts in mutants was altered, with ruptured thylakoid stacking, indistinct stromal lamellae, and some chloroplasts containing white vesicles. These results indicate that chloroplast development in this mutant was affected. As such, changes in the leaf color of mutants could be due to the atypical function and development of plastids in mutant leaves.

### 3.2. Differentially Accumulated Protein Involved in Photosynthesis Metabolism Pathways

In plants, photosynthesis that mainly occurs in chloroplasts is accomplished through a series of reactions that are catalyzed by two photosystems (PSI and PSII) [34]. PSII, the first step in photosynthesis, provides the high redox conditions required for light-dependent reactions. PSII includes more than 20 subunits containing PsbA, PsbO, and PsbQ [35]. PS I catalyze the transfer of electrons, due to light, to stromal ferredoxin from cytochrome b6/f. Several subunits make up PS I, including PsaA, PsaF, PsaG, and PsaL [36]. In higher plants, PS I, PS II, and LHC antennas convert solar energy into chemical energy [37]. In our study, 5 DAPs (BpPsbA, BpPsbB, BpPsbC, BpPsbD, and BpPsbR) were involved in the PS II reaction center, while 8 DAPs (BpPsaA, BpPsaB, BpPsaD, BpPsaF, BpPsaG, BpPsaH, BpPsaL, and BpPsaK) that were related to PS I showed reduced protein abundance in the mutant leaves. Furthermore, 13 DAPs were identified in the LHC, and their abundance was sharply reduced in the mutant leaves. Compared with green leaves, the DAPs involved in photosynthesis experienced different changes in their mutant leaves, indicating that these photosynthetic enzymes (Figure 7) could play an important role in the color mutation of mutant leaves.

The repair process of the photosystem and chloroplast in mutant leaves is reflected by ribosomal proteins. Ribosomal proteins play important roles in the translation of key proteins involved in chloroplast development and photosynthesis and are important components of the protein synthesis machinery [38]. In our study, six ribosomal proteins markedly improved the regeneration of photosynthetic and chloroplast proteins, which would contribute to the repair of the photosystem in mutant leaves. Additionally, proteins with high abundance in the ribosomal cycling factors, DNA repair, and elongation factor Ts would all contribute to chloroplast biogenesis recovery [39]. Substantial increases in protein abundance in the network of molecular chaperones suggested that protein protection was particularly required within the chloroplasts of mutant leaves because HSPs help maintain nascent protein transport and folding. HSP70 could facilitate the transfer of proteins promoting protein translocation into the stroma, which would support both the thylakoid and photosystem II membranes found in chloroplasts [40], and which could require maintaining the function of the damaged photosystems of mutant leaves. Thus, the abundance upregulation of so many HSPs in mutant leaves could reflect measures to prevent plant cell damage and reconstruct cellular homeostasis. After accounting for ribosomal proteins with the up-regulated abundance mentioned above, the assemblies of new protein complexes of PSII and PSI were enhanced in mutant leaves. This would improve the regenerative capacity of chloroplasts and ensure the growth of mutant leaves.

## 4. Materials and Methods

### 4.1. Plant Materials

The golden-yellow hybrid paper mulberry is a bud mutant, and its leaves are golden-yellow throughout the growth period. This study used green leaves and golden-yellow leaves with the same genetic background at the same developmental stages, which grew under natural conditions in the Beijing Botanical Garden (39°48′ N, 116°28′ E). Leaves were harvested from the same position (fifth-leaf samples) in golden-yellow and green hybrid paper mulberry in summer and were immediately frozen in liquid nitrogen and stored at −70 °C for further protein and RNA extraction. Three independent biological replicates were performed.

### 4.2. Determination of Chemical Composition

Pigments were measured according to the method described by Arnon [41] with some modifications. Approximately 0.2 g of leaves from green and mutant hybrid paper mulberry plants were sliced and submerged in 80% acetone to extract the photosynthetic pigments. The supernatant was collected after centrifugation at 10,000 rpm for 15 min. The absorbance was recorded at 665, 649, and 470 nm on a UV-1800 spectrophotometer (Shimadzu Corporation, Kyoto, Japan). We measured the content of chlorophyll intermediaries to investigate physiological changes in the golden-yellow hybrid paper mulberry. The 5-aminolevulinic acid (ALA) content was measured according to the methods described by Dei [42]. Porphobilinogen (PBG), uroporphyrinogen III (Urogen III), and coproporphyrinogen III (Coprogen III) were measured as described by Bogorad [43]. Protoporphyrin IX (Proto IX), Mg-protoporphyrin IX (Mg-Proto IX), and protochlorophyllide (Pchlide) were extracted according to Rebeiz et al. [44]. Carotenoid components were measured using ultra-high performance supercritical fluid chromatography-mass spectrometry (UHPSFC-MS) techniques [45]. The flavonoid content in leaves was measured using a previously reported spectrophotometer method [46]. The reaction system included 1000 μL double distilled water, 100 μL sample, and 500 μL AlCl_3_ (2% v/*v*) reagent. The absorbance of the sample was read at 430 nm after 15 min at room temperature. We established a calibration curve (Y = 1.1847X + 0.0065, r^2^ = 0.9996) with rutin as a standard, and the results were presented as the milligrams of rutin equivalent per g dry weight (mg g^−1^ DW) to obtain flavonoid contents. Car and Chl precursor concentrations were set to 1 in the green leaves. Car and Chl precursor relative values were considered to be the fold changes relative to the green leaves in the mutant leaves.

### 4.3. Transmission Electron Microscopic Analysis

The semi-thin sections and ultrastructure of the chloroplast were observed using mature leaves. The small sections (about 1 mm^2^) were cut from green leaves and golden-yellow mutant leaves. Each sample was fixed at 4 °C overnight in phosphate buffer (0.1 M, pH 7.0) using 3% glutaraldehyde, after which it was incubated with 1% OsO_4_ for 4 h in phosphate buffer and washed three times in phosphate buffer (0.1 M, pH 7.2) for 30 min each time. Samples were then dehydrated using a graded ethanol series for approximately 15 min each (30, 50, 70, 80, 90, 95, and 100%). Following dehydration, the samples were placed in LR White resin (London Resin Company Ltd., London, UK). A Leica EM UC7 microtome (Leica Microsystems IR GmbH, Wetzlar, Germany) and a diamond or glass knife were used to cut thin and semi-thin sections. For light microscopy, semi-thin sections were stained with toluidine blue (Sigma-Aldrich, St. Louis, MO, USA), after which the samples were observed under an optical microscope. For electron microscopy, ultrathin sections were stained with uranyl acetate and lead citrate, and a transmission electron microscope (TEM, Hitachi Ltd., Tokyo, Japan) was used to observe the samples. Three independent experiments were performed for each sample.

### 4.4. Photosynthetic Parameter Measurements

The fifth leaf of golden-yellow mutants and green paper mulberry plants was subjected to photosynthetic parameter analysis. A Li-6400 photosynthesis system was used to assess the transpiration rate (Trmmol), stomatal conductance (Cond), intercellular CO_2_ concentration (Ci), and net photosynthesis (Pn). Chlorophyll fluorescence parameters were measured using a MAXI-IMAGING-PAM chlorophyll fluorometer (Heinz Walz, Effeltrich, Germany). The maximum quantum yield of PS II (Fv/Fm) was determined after the leaves were adapted to the dark for 30 min. Fluorescence parameters were calculated based on the methods described by Xu [47] as follows: maximum quantum yield of PSII, Fv/Fm = (Fm − Fo)/Fm. Fo and Fm are the minimum and maximum fluorescence measured after 30 min of dark adaptation. All measurements were conducted between 08:00 a.m. and 12:00 a.m. Leaf temperature was adjusted to 25 °C during measurements.

### 4.5. Protein Extraction, Digesting, and iTRAQ Labeling

Briefly, 0.5 g leaf samples were homogenized in 2 mL lysis buffer, including 8 M Urea, 50 mM Tris-HCl (pH 8), and 0.2% SDS sonicated for five minutes on ice. The samples were then centrifuged at 12,000× *g* at 4 °C for 15 min, and the supernatant was moved to a new tube. The concentration of protein was determined using a Bradford protein assay. Extracts from each sample were reduced with 2 mM DTT for 1 h and alkylated with sufficient iodoacetic acetate for 1 h in the dark at room temperature. A 4-fold volume of precooled acetone was mixed with the samples and incubated at −20 °C for 1 h; the samples were then centrifuged to collect precipitation. After they were washed three times with cold acetone, the pellets were dissolved in lysis buffer containing 0.1 M triethyl ammonium bicarbonate (TEAB, pH 8.5) and 8 M urea. An equal number of proteins were digested with trypsin (Promega, Madison, WI, USA) at a ratio of 1:50 (w:w) for 16 h at 37 °C and iTRAQ reagents kit (Applied Biosystems, Framingham, MA, USA) were used to label 100 micrograms of digested proteins according to the manufacturer’s protocol.

### 4.6. HPLC Fractionation and LC-MS/MS Analysis

A Rigol L-3000 HPLC system equipped with a C18 column (Waters BEH C18 4.6 × 250 mm, 5 µm) was used. Eluent A was 2% acetonitrile, pH was adjusted to 10.0 using ammonium hydroxide; eluent B was 98% acetonitrile, the pH was adjusted to 10.0 using ammonium hydroxide. Gradient elution: 3% B, 5 min; 3–8% B, 0.1 min; 8–18% B, 11.9 min; 18–32% B, 11 min; 32–45% B, 7 min; 45–80% B, 3 min; 80% B, 5 min; 80–5%, 0.1 min, 5% B, 6.9 min. The iTRAQ-labeled peptide mixtures were eluted at a flow rate of 1 mL min^−1^. During elution, the absorbance was detected at 214 nm, and each fraction was collected every 1 min. The eluted peptide was condensed into 15 components, vacuum-dried, and resynthesized in 0.1% (v/v) formic acid for subsequent analysis. Data processing and analysis were managed using a UHPLC system (Thermo Fisher Scientific, Waltham, MA, USA) equipped with a mass spectrometer (Thermo Fisher Scientific). Peptides were then separated onto a C18 column (15 cm × 150 µm, 1.9 µm) using a 5–100% linear gradient of eluent B (0.1% formic acid in 80% acetonitrile) and eluent A (0.1% formic acid in water) at a flow rate of 600 nL min^−1^. Gradient elution: 5–10% B, 2 min; 10–30% B, 49 min; 30–50% B, 2 min; 50–90% B, 2 min; 90–100% B, 5 min. LC-MS/MS testing was conducted by Novogene Co., Ltd. (Beijing, China). A mass spectrometer in positive polarity mode was used. MS detection parameters were as follows: resolution 60,000 (at 200 m/z), capillary temperature 320 °C, spray voltage 2.3 kV, and scan range m/z 350–1500 Da. From the full MS scan, precursor ions were identified using higher energy fragment collision dissociation fragment analysis at the following settings: 15,000 resolution, max injection time 45 ms, intensity threshold 8.3 × 10^3^, automatic control target value 1 × 10^5^, and normalized collision energy 32%.

### 4.7. Protein Identification and Quantification

MS/MS data were searched using the Proteome Discoverer 2.2 (PD 2.2, Thermo) against the paper mulberry databases, and its protein sequences were identified based on transcriptome and genome sequencing. Search parameters were as follows: a mass tolerance for precursor ion scans and product ion scans were 10 ppm and 0.02 Da, respectively. Carbamidomethyl was used as fixed modifications in PD 2.2. Variable modifications in PD 2.2 include lysine, N-terminus acetylation, methionine oxidation, and iTRAQ 8-plex of tyrosine. In terms of protein identification, at the peptide and protein levels, the false discovery rate (FDR) was less than 1.0%, and proteins were identified with at least one unique peptide, respectively. The indistinguishable protein peptides were grouped into proteomes according to the MS/MS analysis. The quantification of iTRAQ is the reporter quantification (iTRAQ8-plex). To select the differentially accumulated proteins (DAP), a 1.2-fold cutoff and *p* < 0.05 were considered.

### 4.8. Proteomic Data Analysis

Functional analysis of identified proteins was performed using Gene Ontology (http://www.geneontology.org, accessed on 8 May 2021). Differentially accumulated proteins were then placed in the KEGG database (http://www.genome.jp/kegg/pathway.html, accessed on 8 May 2021) or the COG database (http://www.ncbi.nlm.nih.gov/COG/, accessed on 9 May 2021). In order to determine functional subgroups and metabolic pathways in which the differentially accumulated proteins were enriched, GO, and KEGG pathway enrichment analyses were conducted. The cluster analysis of the differentially accumulated proteins was conducted by using Cluster 3.0, and the heatmap was generated using TreeView version 1.6.

### 4.9. RNA Extraction and qRT-PCR Analysis

Total RNA was extracted with a TransZol™ RNA Extraction Kit (TransGen, Beijing, China) from green and mutant leaves according to the manufacturer’s recommendations. The integrity and quantity of each RNA sample were checked using 1.2% agarose gel and assessed with a Nanodrop 2000 (Thermo Fisher, Waltham, MA, USA). SuperScript II reverse transcriptase (Takara, Dalian, China) was used to perform cDNA synthesis according to the manufacturer’s recommendations. The cDNAs were diluted ten-fold for qRT-PCR. Primer 5 program was used to design primer pairs, which are shown in Appendix A. The qRT-PCR was run using the SYBR Premix Ex TaqII kit (Takara, Dalian, China) on a StepOne real-time PCR system (Applied Biosystems, Foster City, CA, USA) according to the manufacturer’s instructions. The qRT-PCR reaction system included 10 μL of SYBR, 1 μL of cDNA, 0.4 μL of ROX, 0.4 μL gene-specific primers, and 7.8 μL of ddH_2_O. The PCR cycles were as follows: 95 °C for 30 s, followed by 35 cycles of 95 °C for 5 s, and 60 °C for 30 s. Relative transcript levels were calculated according to the level of internal control (GAPDH) using the method of the 2^−^^ΔΔCt^ formula. All data were presented as mean ± standard deviation (SD) (*n* = 3).

### 4.10. Statistical Analysis

All assays were conducted in three replications, and the significance was set to determine via one-way analysis of variance (ANOVA), while *p* < 0.05 was considered statistically significant. Results were expressed as mean ± standard deviation.

## 5. Conclusions

We here performed a systematic analysis between green and golden-yellow leaf hybrid paper mulberry using a variety of research methods, including physiological parameter determination, cytological observation, and the iTRAQ-based proteomic-wide MS analysis. We found that the chlorophyll content dramatically decreased and the Car/Chl and Chl a/b ratios increased in golden-yellow leaves compared with the green leaves. These results suggest that the changes in photosynthetic pigment content and ratio may be the reason for the formation of mutant golden-yellow leaves. Further studies showed that the chloroplast structure was defective in golden-yellow leaves. Therefore, the atypical function and development of plastids may be another important reason for the mutant leaf coloring. Next, proteomic analyses of the golden-yellow leaf mutant of hybrid paper mulberry were performed for the first time. In the mutant leaves, 168 proteins with different abundances were found, of which 117 proteins were down-regulated, and 51 proteins were up-regulated. Most of the DAPs were involved in the chlorophyll biosynthesis, photosynthesis, and ribosome pathways. Additionally, these differentially accumulated proteins perform several important functions relating to chloroplast morphogenesis, photosynthetic electron transport, and light absorption. These results further reveal the possible molecular mechanism of leaf color formation in mutants. In summary, this study expanded our understanding of the formation mechanism of the golden-yellow leaf phenotype in hybrid paper mulberry. Furthermore, this will lay the groundwork for further study on the mechanism of leaf coloring in other woody plants.

## Figures and Tables

**Figure 1 ijms-23-00127-f001:**
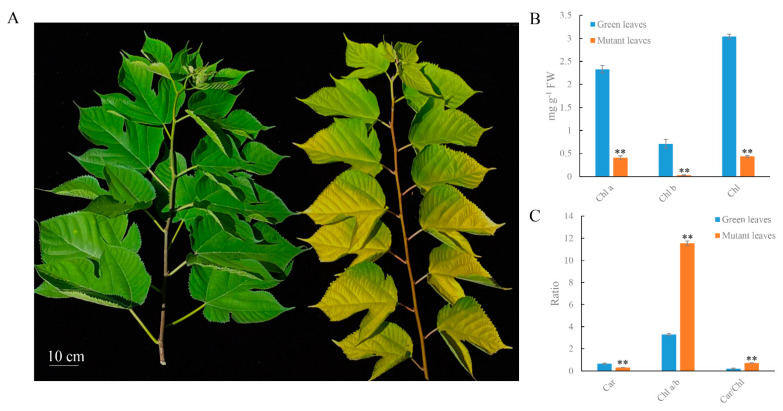
Phenotypic characterization and pigment content (chlorophyll and carotenoid) of green and mutant leaves. (**A**) Phenotypic characterization of green and mutant leaves. (**B**) Chlorophyll (Chl) and carotenoid (Car) content in mg g^−1^ fresh weight. (**C**) The ratio of the Chl a to Chl b and Car to Chl. The values shown are mean ± SD from three independent experiments. Significant differences were determined using Student’s *t*-test in mutant and green leaves (** *p* < 0.01). Bar = 10 cm (**A**).

**Figure 2 ijms-23-00127-f002:**
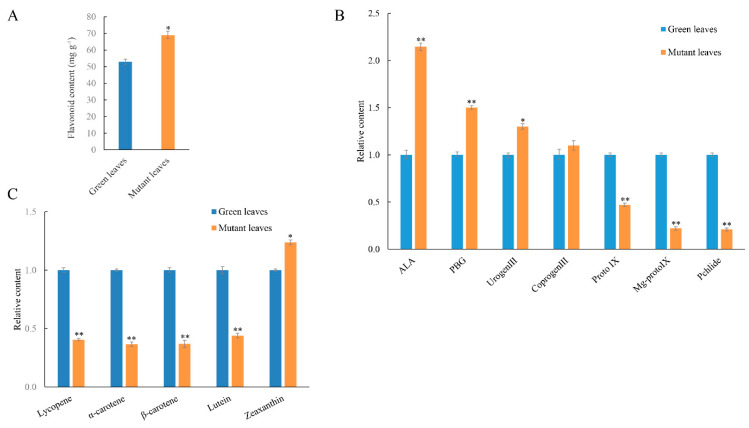
Pigment contents in leaves of hybrid paper mulberry. (**A**) Total content of flavonoids in green and mutant leaves. (**B**) Analysis of relative chlorophyll intermediary contents in green and mutant leaves. (**C**) Relative carotenoid component contents in green and mutant leaves. Error bars indicate mean ± SD, from three independent replicates. Student’s *t*-test was used to identify significant differences in mutant and green leaves (* *p* < 0.05, ** *p* < 0.01). ALA, 5-aminolevulinic acid; PBG, porphobilinogen; Urogen III, uroporphyrinogen III; Coprogen III, coproporphyrinogen III; Proto IX, protoporphyrin IX; Mg-Proto IX, Mg-protoporphyrin IX; Pchlide, protochlorophyllide.

**Figure 3 ijms-23-00127-f003:**
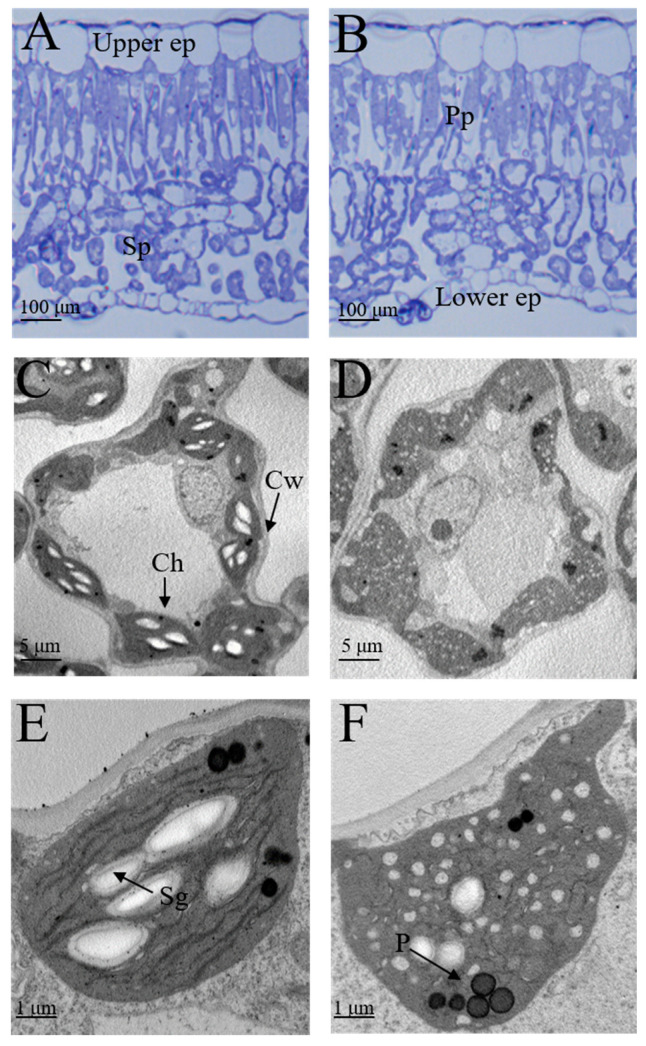
Semi-thin section of a cross-section of the leaves and transmission electron microscopic images of chloroplast in green and mutant leaves. (**A**,**B**) Cross-section of leaves from green and mutant plants. (**C**,**D**) Structure of mesophyll cells in green and mutant leaves. (**E**,**F**) Ultrastructure of the chloroplast of green and mutant leaves. Upper ep, upper epidermis; Lower ep, lower epidermis; Pp, palisade parenchyma; Sp, spongy parenchyma; Ch, chloroplast; Sg, starch granule; Cw, cell wall; P, plastoglobuli. (Scale bars: 100 μm in (**A**,**B**); 5 μm in (**C**,**D**); 1 μm in (**E**,**F**)).

**Figure 4 ijms-23-00127-f004:**
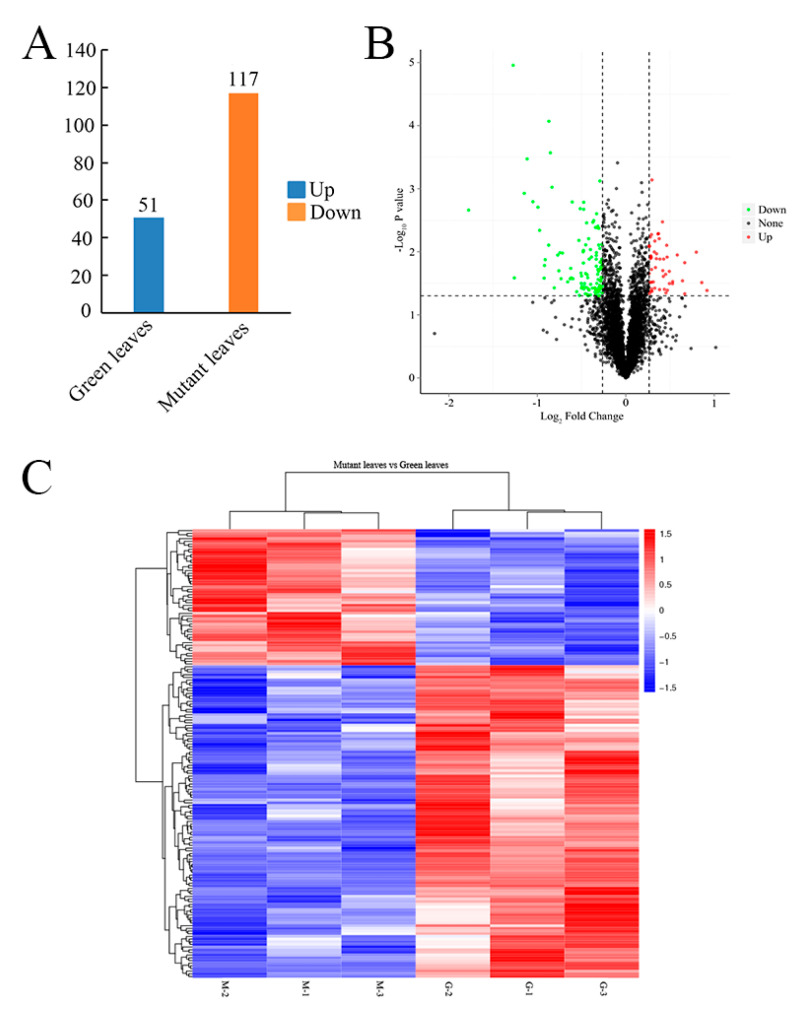
Identified differentially accumulated proteins between green and mutant leaves. (**A**) Volcano plot of all DAPs in mutant compared with green leaves. (**B**) Quantitative analysis of the proteome between green and mutant leaves. In green (down-regulated) and in red (up-regulated): proteins with *p* < 0.05 and |log_2_FC| > 0.26; in black: proteins alone. (**C**) Hierarchical cluster analysis of DAPs. The color spectrum from blue to red indicates protein expression intensity, ranging from low to high, respectively. Green leaves-1, G-1; Green leaves-2, G-2; Green leaves-3, G-3; Mutant leaves-1, M-1; Mutant leaves-2, M-2; and Mutant leaves-3, M-3.

**Figure 5 ijms-23-00127-f005:**
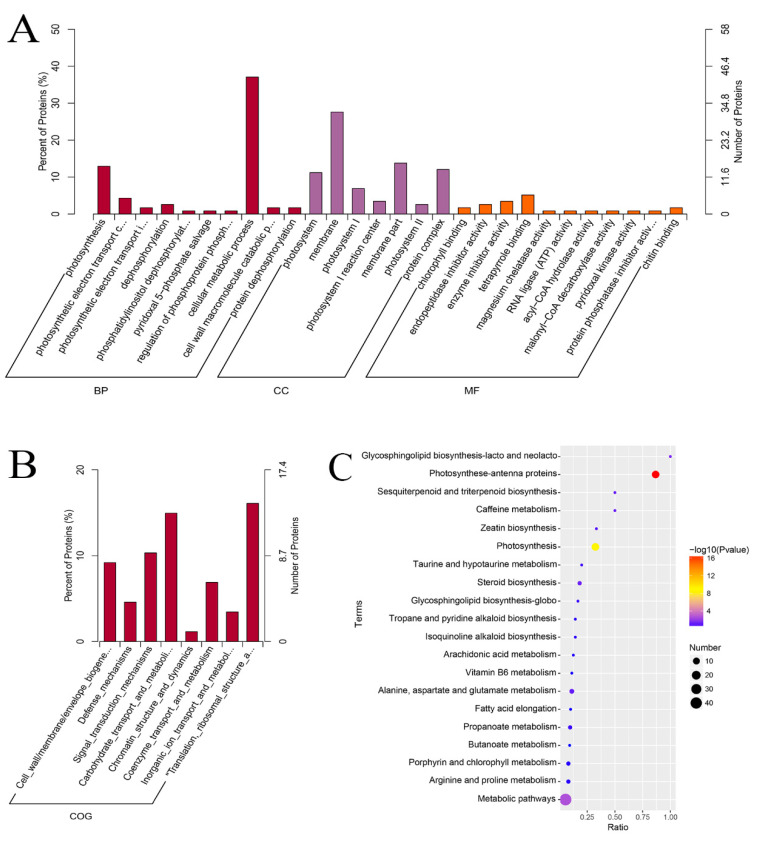
(**A**) Differentially accumulated proteins in green and mutant leaves via gene ontology (GO) classification. (**B**) Green and mutant leaf protein categorization by clusters of orthologous groups (COG). (**C**) Pathway enrichment analysis of DAPs between mutant and green leaves. Nineteen pathways (*p*-value < 0.05) were selected for this analysis.

**Figure 6 ijms-23-00127-f006:**
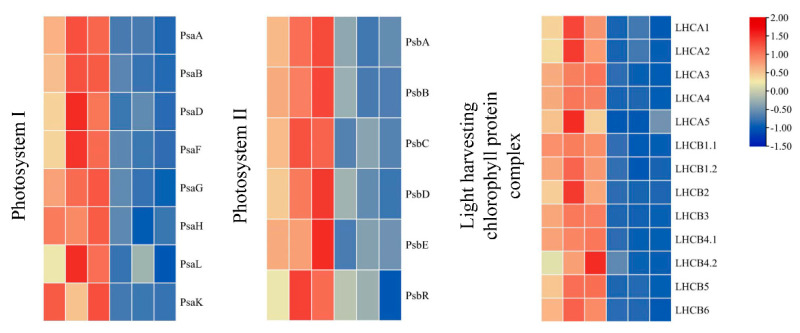
The differentially accumulated proteins associated with photosynthesis in green and mutant leaves. Identified DAPs are shown using a heatmap. Green leaves-1, G-1; Green leaves-2, G-2; Green leaves-3, G-3; Mutant leaves-1, M-1; Mutant leaves-2, M-2; and Mutant leaves-3, M-3 (left to right: G-1, G-2, G-3, M-1, M-2 and M-3).

**Figure 7 ijms-23-00127-f007:**
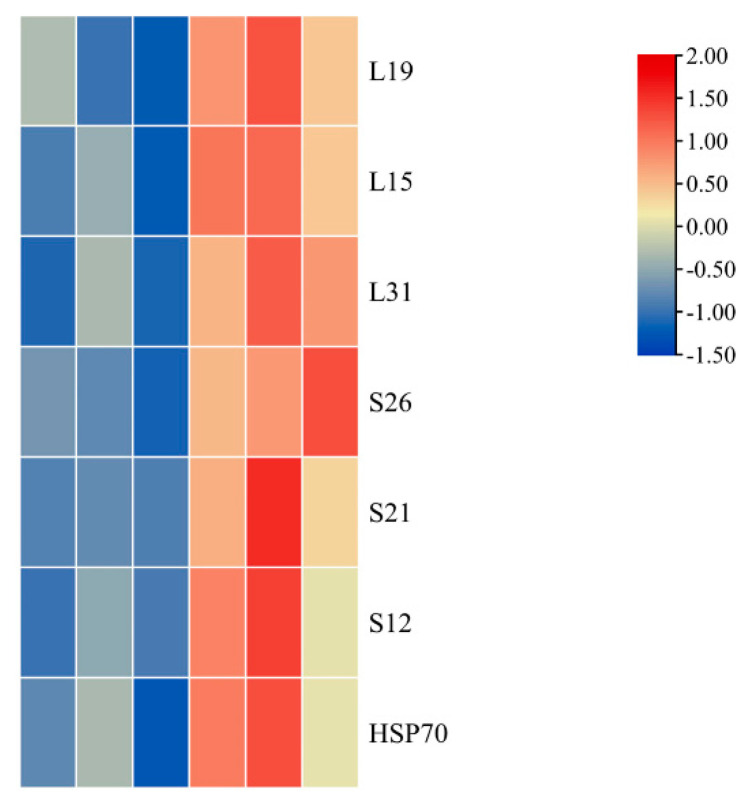
Identified differentially accumulated proteins in ribosome pathways are shown using a heatmap. Green leaves-1, G-1; Green leaves-2, G-2; Green leaves-3, G-3; Mutant leaves-1, M-1; Mutant leaves-2, M-2; and Mutant leaves-3, M-3 (left to right: G-1, G-2, G-3, M-1, M-2, and M-3).

**Figure 8 ijms-23-00127-f008:**
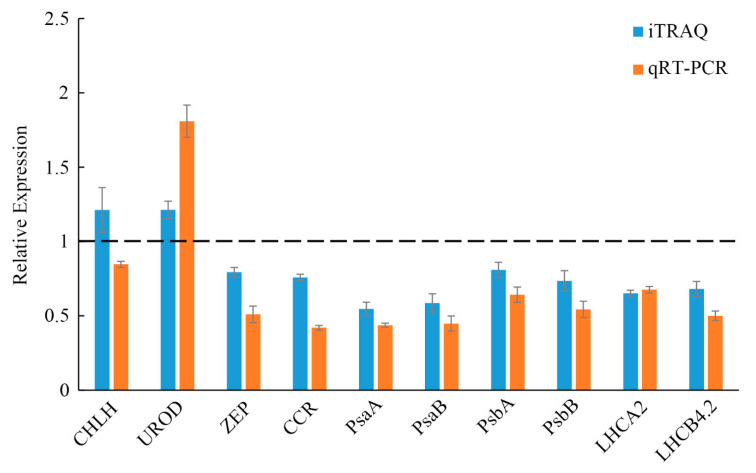
Comparison of levels of RNA and protein of differentially accumulated proteins via qRT-PCR and iTRAQ. Green leaf data was set to 1.0. Displayed values in mutant leaves are means ± SD (*n* = 3). CHLH: magnesium chelatase H; UROD: uroporphyrinogen decarboxylase; ZEP: zeaxanthin epoxidase; CCR: cinnamoyl-CoA reductase; PsaA: photosystem I P700 chlorophyll a apoprotein A1; PsaB: photosystem I P700 chlorophyll a apoprotein A2; PsbA: photosystem II protein D1; PsbB: photosystem II CP47 reaction center protein; LHCA2: chlorophyll a-b binding protein 7; LHCB4.2: chlorophyll a-b binding protein CP29.3.

**Table 1 ijms-23-00127-t001:** Analysis of leaf gas-exchange in mutant and green leaves of the hybrid paper mulberry.

Materials	Net Photosynthetic Rate (μmol·m^−2^·s^−1^)	Stomatal Conductance (mmol·m^−2^·s^−1^)	Intercellular CO_2_ Concentration (μmol·mol^−1^)	Transpiration Rate (mol·m^−2^·s^−1^)
Green leaves	23.55 ± 0.37	367.83 ± 2.28	245.5 ± 2.67	6.91 ± 0.79
Mutant leaves	14.81 ± 0.08 *	303.33 ± 1.31	315.1 ± 1.84 *	7.45 ± 0.14

(*: *p* < 0.05).

**Table 2 ijms-23-00127-t002:** Chlorophyll fluorescence kinetic parameters of green and mutant leaves.

Materials	Fo	Fm	Fv	Fv/Fm
Green leaves	517.83 ± 13.5	2594.83 ± 26.5	2077.00 ± 25.9	0.80 ± 0.02
Mutant leaves	464.63 ± 6.43	1243.00 ± 9.44 *	778.38 ± 8.14 **	0.58 ± 0.04 *

Error bars represent mean ± SD based on three biological replicates (*: *p* < 0.05, **: *p* < 0.01).

## Data Availability

Not applicable.

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
