# Peer review of "iTRAQ-Based Quantitative Proteomics Analysis Reveals the Mechanism of Golden-Yellow Leaf Mutant in Hybrid Paper Mulberry"

_ijms, 2021, doi:10.3390/ijms23010127_

Round 1
Reviewer 1 Report
Authors investigated physiological and cytological parameters supported via iTRAQ-based proteomic to uncover the mechanisms of the golden-yellow phenotype of the paper mulberry plant. Authors compared green and golden-yellow leaves of hybrid paper mulberry. Authors revealed the the wild type and mutant differs in pigments level, structure of chloroplasts and expression of proteins related to chlorophyll synthesis, carotenoid metabolism, and photosynthesis. The manuscript contains interesting results, however their presentation needs improvements before publication.
Major concerns:
1. Repetitions 1-2-3 shows a great difference both in G and M samples as shown in three heat maps.
2. Results of qRT-PCR as comparison between green and mutant leaves given in fold change units must be presented and must replace Figure 8.
3. Compared levels of RNA and protein (Figure 8) are similar (by overlapig error bars) only in two from ten analyzed genes/proteins indicating that transcriptomic and proteomic analyses are very different.
4. Figure preparation:
Figure 2 - explian in the figure's caption what does the relative content means?
Figure 5C, the font of listed pathwas seems to be too small
Figure 7A is not suitable in results because it probably shows a literature data, should be removed at all.
5. Material. It has to be specified at which developemntal stage the analysed leaves were, each time leaf number one or leaf number three or other?
5. Conclusion is highly descriptive, must be rewritten.
Editorial mistakes:
Line 71, inset a reference number after "Liu et al."
Line 97 authors wrote "(Figure 1)" and further use "(Fig. 1B)." in line 103 or "(Fig. 1C)." in line 105, please unify the format of figure's citation in the text
line 121 ' correct the format of "mg g -1"
line 243, consde rchanging "compare" to " as compared" or "in comparison with"
line 348, transfer the subheading "3.2. Differentially Accumulated Protein Involved in Photosynthesis Metabolism Pathways" to the next line
line 371, explain "ribosomal cycle factor"
line 420, write "4 °C" as in line 448
Reviewer 2 Report
The manuscript is considerably improved. The authors raised all of my requirements and suggestions; thus, now I have no objections to the manuscript.
I have only one minor suggestion. Because all figures must be self-explanatory, denotations G-1…M-3 should be explained not only below figure 4 but also below Figures 6 and 7.
Enter is missing in line 348 (Discussion). The heading of subsection 3.2 must be distinguished and must be written in italics.
Round 2
Reviewer 1 Report
The manuscript was improved in accordance to my suggestions and might be published.
Author Response
Thank you very much for all your help.These comments are all valuable and very helpful for revising and improving our paper, as well as the important guiding significance to our future researches.
This manuscript is a resubmission of an earlier submission. The following is a list of the peer review reports and author responses from that submission.
Round 1
Reviewer 1 Report
Authors compared the golden-yellow mutant leaves with normal green leaves in Hybrid Paper Mulberry in terms of chloroplast structure, pigment content, photosynthetic properties, chlorophyll and carotenoid synthesis precursors and investigated the changes in proteome.
Major points:
- Lines 14-16 “Compared with wild type plants, the mutants showed golden-yellow leaves, reduced chlorophyll and carotenoid content, increased flavonoid content, fewer thylakoid stacks and retarded growth.”
Pease specify whether this description relates to previous published articles. Whether yes, this manuscript is not novel to science.
- Starting from line 16, abstract describes only proteomic data, whereas physiological, cytological results are neglected. Please rewrite.
- Confusing informations in lines 57-65. Reference 18 is to paper mulberry (Broussonetia papyrifera), not the hybrid (Broussonetia kazinoki × Broussonetia papyrifera), therefore the use of hybrid as forage is probably not true, the biomass obtained from a species with golden-yellow leaves and reduced photosynthesis rates would be probably not sufficient. Reference 16 relates to phosphoroteomic analyses of paper mulberry and (not mentioned that it is the the hybrid), and therefore do not present the relevance of the hybrid. Reference 17 also refers to paper mulberry, not the hybrid, therefore there is no economic significance of the hybrid described in the Introduction. Only the reference 15 relates to the hybrid Broussonetia kazinoki × Broussonetia papyrifera, but authors name it paper mulberry, not the hybrid. Therefore the whole paragraph (lines 57-65) must be rewritten to clearly present data related to the Broussonetia papyrifera and the Broussonetia kazinoki × Broussonetia papyrifera and define what exactly is known about the golden-yellow hybrid.
- Misleading results. Authors stated that “the mutant leaves exhibited golden-yellow color because of high Car accumulation rate” (lines (108-109). However, Figure 2c shows that only one from five analyzed Car was higher in the mutant and Table 1 shows that mutant display halved car levels. Further in line 132, Authors wrote that “Due to high rates of Car accumulation, we measured the major carotenoid components in our samples”. Please clarify the statement.
- Specify how “relative content” was calculated in Figure 2b,c.
- LogFC threshold is arbitrary, but the vaue at least 1 should be kept. Authors specified that “ratio of protein abundance > 1.2/0.8-fold)” was used. What was the logFC for downregulated proteins? line 228 Authors wrote: “downregulated): proteins with p < 0.05 and fold change < 0.83”. Figure 4B shows that Log2FC 0.25 was used at the X axis. Please explain and use the same cutoff (higher than 1) for up- and down-regulated proteins.
- Lines 236-248 are a part of Introduction in Results. Move it to Introduction and keep a brief one-sentenced description to explain the necessity/importance of described results.
- Figures. Figure5C font size is too small. Figure 6, the scheme do not include “tetrapyrrole-binding protein (GUN4, Bp13g0174)” described in the text. Figure 7A is not necessary because does not include “Schematic representation of DAPs associated with photosynthesis” as specified in this figure caption.
- Methodological issue. “G-1, G-2, G-3, M-1, M-2 and M-3” should refer to three biological repetitions of green and mutant samples. However, Authors stated that 1 refers to bud, 2 refers to young leaf and 3 refers to mature leaf. In my opinion, all of the three selected developmental stages of leaves should be analyzed in triplicates and described as separate stages since Figure 8 shows a great differences in protein abundances between 1, 2 and 3.
- Discussion should be rewritten to highlight novel results.
- Conclusion is a simple summary o f results and not conlusion. Rewrite it and change order of sentence 1 with 2.
Minor points:
Line 6, accept changes indicated with track changes format
Line 27, consider adding to keywords the Latin name of the analyzed species
Lines 103, 104, the abbreviations were not explained at first use
Lines 103, 104, unify the writing, “Chlorophyll” or “chlorophyll”
Line 148, Sp in the figure and sp in the textare used
Line 205 use the abbreviation that was introduced before
Line 267, abbreviations were explained already in the text
Line 286, define G-1, G-2, G-3, M-1, M-2 and M-3, also in methods
Line 191, explain “RPs”
Line 343, authors write about protein subunits and italicize their names as for genes
Many mistakes in Reference list, not italicized Latin names, abbreviations and full journal names are used. Please correct.
Author Response
Reviewer 1
Comments and Suggestions for Authors
Major points:
- Lines 14-16 “Compared with wild type plants, the mutants showed golden-yellow leaves, reduced chlorophyll and carotenoid content, increased flavonoid content, fewer thylakoid stacks and retarded growth.” Please specify whether this description relates to previous published articles. Whether yes, this manuscript is not novel to science.
Response: Thanks very much for your question. Most of the previous studies focused on the effect of chlorophyll and carotenoid content on leaf color mutants. In our study, the results showed that the flavonoid content was increased except for the change of pigment content in golden-yellow mutant. Modern pharmacological studies have shown that the flavonoids are beneficial in drug development and health care, such as anti-cancer, anti-inflammation, anti-bacteria, and other pharmacological activities, so our study provides a direction for the study of leaf color mutants in the future. On the other hand, the golden-yellow hybrid paper mulberry with high flavonoid content will have better feeding value, and this has important theoretical and application value for feed application of hybrid paper mulberry.
- Starting from line 16, abstract describes only proteomic data, whereas physiological, cytological results are neglected. Please rewrite.
Response: Thanks very much for your question. We have modified the article according to your suggestion.
- Confusing informations in lines 57-65. Reference 18 is to paper mulberry (Broussonetia papyrifera), not the hybrid (Broussonetia kazinoki × Broussonetia papyrifera), therefore the use of hybrid as forage is probably not true, the biomass obtained from a species with golden-yellow leaves and reduced photosynthesis rates would be probably not sufficient. Reference 16 relates to phosphoroteomic analyses of paper mulberry and (not mentioned that it is the phosphoremia hybrid), and therefore do not present the relevance of the hybrid. Reference 17 also refers to paper mulberry, not the hybrid, therefore there is no economic significance of the hybrid described in the Introduction. Only the reference 15 relates to the hybrid Broussonetia kazinoki × Broussonetia papyrifera, but authors name it paper mulberry, not the hybrid. Therefore, the whole paragraph (lines 57-65) must be rewritten to clearly present data related to the Broussonetia papyriferaand the Broussonetia kazinoki × Broussonetia papyriferaand define what exactly is known about the golden-yellow hybrid.
Response: Thank you very much for your indicating the valuable problem, which are very helpful for the improvement of our work. Although the title of reference 16,17 and 18 was wrote as the paper mulberry (Broussonetia papyrifera), the main text explained that it was the hybrid paper mulberry obtained by hybridization (Broussonetia kazinoki × Broussonetia papyrifera).
- Authors stated that “the mutant leaves exhibited golden-yellow color because of high Car accumulation rate” (lines (108-109). However, Figure 2c shows that only one from five analyzed Car was higher in the mutant and Table 1 shows that mutant display halved car levels. Further in line 132, Authors wrote that “Due to high rates of Car accumulation, we measured the major carotenoid components in our samples”. Please clarify the statement.
Response: Thank you for pointing out this problem. Due to high rates of Car/Chl, we measured the vital carotenoid components in hybrid paper mulberry.
- Specify how “relative content” was calculated in Figure 2b, c.
Response: Thank you for pointing out this problem. Relative content refers to the content ratio of mutant leaves to green leaves, taking the content of green leaves as 1.
- LogFC threshold is arbitrary, but the value at least 1 should be kept. Authors specified that “ratio of protein abundance > 1.2/0.8-fold)” was used. What was the logFC for downregulated proteins? line 228 Authors wrote: “downregulated): proteins with p < 0.05 and fold change < 0.83”. Figure 4B shows that Log2FC 0.25 was used at the X axis. Please explain and use the same cutoff (higher than 1) for up- and down-regulated proteins.
Response: Thank you very much for your indicating the valuable problem, which are very helpful for the improvement of our work. In this study, 168 proteins showed significant difference (|log2FC| > 0.26; FC > 1.2 or FC < 0.83; [fold change, FC]; p < 0.05) between mutant and green leaves, including 51 up-regulated and 117 down-regulated proteins.
- Lines 236-248 are a part of Introduction in Results. Move it to Introduction and keep a brief one-sentenced description to explain the necessity/importance of described results.
Response: Thanks very much for your question. We have modified the article according to your suggestion.
- Figure5C font size is too small. Figure 6, the scheme does not include “tetrapyrrole-binding protein (GUN4, Bp13g0174)” described in the text. Figure 7A is not necessary because does not include “Schematic representation of DAPs associated with photosynthesis” as specified in this figure caption.
Response: Thanks very much for your kind reminding. Figure 5C, figure 6 and figure 7A were revised, please check it.
- Methodological issue. “G-1, G-2, G-3, M-1, M-2 and M-3” should refer to three biological repetitions of green and mutant samples. However, Authors stated that 1 refers to bud, 2 refers to young leaf and 3 refers to mature leaf. In my opinion, all of the three selected developmental stages of leaves should be analyzed in triplicates and described as separate stages since Figure 8 shows a great difference in protein abundances between 1, 2 and 3.
Response: Thank you for pointing out this problem. The leaves of the same position of golden-yellow and green hybrid paper mulberry were harvested in summer, which were one bud, the second leaf (young leaf) and the fifth leaf (mature leaf) respectively. The mature leaf samples (“Green leaves-1, G-1; Green leaves-2, G-2; Green leaves-3, G-3; Mutant leaves-1, M-1; Mutant leaves-2, M-2; and Mutant leaves-3, M-3”) were used for proteomic analyses. On the other hand, one bud, the young leaf and the mature leaf samples were used for RNA extraction and qRT-PCR analysis. Three independent biological replicates were acquired.
- Discussion should be rewritten to highlight novel results.
Response: Thanks very much for your suggestion. We supplemented the scientific discussion with the novel results in pages 19-21.
- Conclusion is a simple summary of results and not conclusion. Rewrite it and change order of sentence 1 with 2.
Response: Thanks very much for your suggestion. We rewrite the conclusion in page 25.
Minor points:
- Line 6, accept changes indicated with track changes format
Response: Thanks very much for your corrections. The corresponding revision has been made in the manuscript, and marked in red color.
- Line 27, consider adding to keywords the Latin name of the analyzed species
Response: Thanks very much for your suggestion. The Latin name have been added in the manuscript. Please check line 60.
- Lines 103, 104, the abbreviations were not explained at first use
Response: Thanks very much for your suggestion. The abbreviations have been explained in the manuscript. Please check line 106.
- Lines 103, 104, unify the writing, “Chlorophyll” or “chlorophyll”
Response: Thanks very much for your question. We have modified the article according to your suggestion.
- Line 148, Sp in the figure and sp in the text are used
Response: Thanks very much for your question. We have modified the article and figure according to your suggestion.
- Line 205 use the abbreviation that was introduced before
Response: Thanks very much for your question. We have modified the article according to your suggestion. Please check it.
- Line 267, abbreviations were explained already in the text
Response: Thanks very much for your question. We have modified the article according to your suggestion. Please check it.
- Line 286, define G-1, G-2, G-3, M-1, M-2 and M-3, also in methods
Response: We really appreciate this advice, and we indicate the full names of the abbreviation “M-1 (mutant leaves-1), M-2 (mutant leaves-2), M-3 (mutant leaves-3), G-1 (green leaves-1), G-2 (green leaves-2) and G-3 (green leaves-3),” in lines 248-249.
- Line 191, explain “RPs”
Response: Thanks very much for your question. RPs: ribosomal proteins. Please check line 291.
- Line 343, authors write about protein subunits and italicize their names as for genes
Response: Thanks very much for your question. We have modified the article according to your suggestion. Please check it.
- Many mistakes in Reference list, not italicized Latin names, abbreviations and full journal names are used. Please correct.
Response: Thanks very much for your question. We have modified the article according to your suggestion.

Reviewer 2 Report
The manuscript contains interesting results and can be considered for publication. However, it must be considerably improved because it includes mistakes. My main reservation concern the section Results. Authors must carefully read, check and correct the entire section and remove mistakes.
- Lines 107-108. There is written that the content of flavonoid is expressed in mg/g, but the content in Figure 2A is presented in percentages, not in mg/g. Additionally, it is not known in percentages of what. Moreover, firstly, it should be explained what the ‘g’ means. Does it mean fresh weight or maybe a dry matter of leaves? Secondly, instead of mg/g, it should be mg g-1 (for example, mg g-1 FW).
- In my opinion, data presented in Table 1 may be presented on the graph, and such a graph may be included in Figure 1.
- Line 129. It is written that the increase in the content of some intermediates of the chlorophyll biosynthesis pathway was 10-20%, but in Figure 1B, we can see that the rise in ALA content was over 100%.
- Figure 3. Arrows indicating CW, Sg, and P must be precisely placed on the pictures. Moreover, the abbreviation and the arrow indicating thylakoids should be added because such abbreviation is used in line 154.
- Table 2 and 3. Similarly, like in the case of Table 1, authors should consider changing them into graphs. Moreover, below Table 3, all abbreviations must be explained.
- Figure 4. It is not clear what the denotations M_1 to M_3 and G_1 to G_3 mean. It is not explained nither below the figure nor in Materials and methods. It may be a bigger problem because I am confused reading subsection 2.9. Why was the PCR made for three developmental stages - bud, young leaf, and mature leaf? Dose denotations M_1 to M_3 and G_1 to G_3 mean bud, young leaf, and mature leaf? It must be explained.
- Line 253. Figure 6A does not show that the increase was by 1.20 and 1.24-fold. Figure 6 contains only some red-blue bars, and it is not known what these bars mean. Moreover, the caption of Figure 6 should be improved because the figure presents a schematic representation of two pathways, not DAPs. Or maybe it shows DAPs, but it must be precisely written how.
- Figure 7A. This part of Figure 7 must be removed from the manuscript, or it may be alternatively moved to the introduction. There is written in the legend that this part is a “Schematic representation of DAPs associated with photosynthesis.”, but where are the DAPs marked? It is a general scheme of photosynthesis, and such a scheme should not be placed in the Results. If yes, all DAPs should be shown, and all abbreviations and denotations should be explained in the figure legend. Maybe some simplified scheme will be better here?
Minor points
- The Latin name must be added to the title, abstract, or introduction. The abbreviation “B.” used in line 57 may not be commonly known.
- Keywords should not be the same as words already used in a title.
- “Differentially accumulated proteins”, “Protein differences”, “Differential accumulated proteins”, and “DAPs” are used exchangeably. It should be unified and corrected.
- Figure 8 should be smaller.
- Line 313-314. The sentence “These findings indicated that the molecular mechanism of the golden-yellow mutant was complicated” does not sound scientifically good, and it looks like a response of a student to the tricky question because, after all, everything is complicated. (Please excuse this joke.)
- Line 424. The sentence is incorrect because ALA content cannot be extracted. Extracted can be ALA, and its content may be determined.
Author Response
Reviewer 2
Comments and Suggestions for Authors
The manuscript contains interesting results and can be considered for publication. However, it must be considerably improved because it includes mistakes. My main reservation concerns the section Results. Authors must carefully read, check and correct the entire section and remove mistakes.
Response: We sincerely appreciate your recognition of our work. Thank you very much. We have carefully made some revisions to further improve our manuscript. A revised manuscript with the correction sections marked with the red color was resubmitted for easy check/editing purpose.
- Lines 107-108. There is written that the content of flavonoid is expressed in mg/g, but the content in Figure 2A is presented in percentages, not in mg/g. Additionally, it is not known in percentages of what. Moreover, firstly, it should be explained what the ‘g’ means. Does it mean fresh weight or maybe a dry matter of leaves? Secondly, instead of mg/g, it should be mg g-1 (for example, mg g-1 FW).
Response: Thank you for pointing out this problem. We have modified the Figure 2A and the results were expressed in the form of milligrams of rutin equivalent per g dry weight (mg g-1 DW).
- In my opinion, data presented in Table 1 may be presented on the graph, and such a graph may be included in Figure 1.
Response: Thanks very much for your question. Because of the high content of chlorophyll and low carotenoids, the two values differ greatly, and we made the corresponding graph and found that the result shown in the graph is not as obvious as that shown in the table, so we show the results with the table.
- Line 129. It is written that the increase in the content of some intermediates of the chlorophyll biosynthesis pathway was 10-20%, but in Figure 1B, we can see that the rise in ALA content was over 100%.
Response: Thank you very much for pointing out this mistake. We have corrected the sentences “Further detailed analysis showed that the chlorophyll synthesis precursor 5-aminolevulinic acid (ALA), porphobilinogen (PBG), uroporphyrin III (Urogen III) and coproporphyrin III (Coprogen III) contents increased significantly in the mutant leaves. On the other hand, the magnesium protoporphyrin (Mg-Proto), protoporphyrin IX (Proto IX) and protochlorophyllide (Pchlide) contents decreased markedly in the mutant leaves compared with green leaves.” in page 4.
- Figure 3. Arrows indicating CW, Sg, and P must be precisely placed on the pictures. Moreover, the abbreviation and the arrow indicating thylakoids should be added because such abbreviation is used in line 154.
Response: Thank you very much for your comments. We have modified the Figure 3 and deleted the abbreviation of thylakoids according to your suggestion.
- Table 2 and 3. Similarly, like in the case of Table 1, authors should consider changing them into graphs. Moreover, below Table 3, all abbreviations must be explained.
Response: Thanks very much for your question. We made the corresponding graph and found that the result shown in the graph is not as obvious as that shown in the table, so we showed the results with the table. We have explained all abbreviations according to your suggestion.
- Figure 4. It is not clear what the denotations M_1 to M_3 and G_1 to G_3 mean. It is not explained below the figure nor in Materials and methods. It may be a bigger problem because I am confused reading subsection 2.9. Why was the PCR made for three developmental stages - bud, young leaf, and mature leaf? Dose denotations M_1 to M_3 and G_1 to G_3 mean bud, young leaf, and mature leaf? It must be explained.
Response: Thank you for pointing out this problem. The leaves of the same position of golden-yellow and green hybrid paper mulberry were harvested in summer, which were one bud, the second leaf (young leaf) and the fifth leaf (mature leaf) respectively. The mature leaf samples (“Green leaves-1, G-1; Green leaves-2, G-2; Green leaves-3, G-3; Mutant leaves-1, M-1; Mutant leaves-2, M-2; and Mutant leaves-3, M-3”) were used for proteomic analyses. On the other hand, one bud, the young leaf and the mature leaf samples were used for RNA extraction and qRT-PCR analysis. The related description for M_1 to M_3 and G_1 to G_3 has been added on lines 248-249.
- Line 253. Figure 6A does not show that the increase was by 1.20 and 1.24-fold. Figure 6 contains only some red-blue bars, and it is not known what these bars mean. Moreover, the caption of Figure 6 should be improved because the figure presents a schematic representation of two pathways, not DAPs. Or maybe it shows DAPs, but it must be precisely written how.
Response: Thank you very much for your comments. We have modified the Figure 6 according to your suggestion. “Identified differentially accumulated proteins associated with pigment biosynthesis between green leaves and mutant leaves”.
- Figure 7A. This part of Figure 7 must be removed from the manuscript, or it may be alternatively moved to the introduction. There is written in the legend that this part is a “Schematic representation of DAPs associated with photosynthesis.”, but where are the DAPs marked? It is a general scheme of photosynthesis, and such a scheme should not be placed in the Results. If yes, all DAPs should be shown, and all abbreviations and denotations should be explained in the figure legend. Maybe some simplified scheme will be better here?
Response: Thank you very much for your comments. We have modified the Figure 7 according to your suggestion.
Minor points
- The Latin name must be added to the title, abstract, or introduction. The abbreviation “B.” used in line 57 may not be commonly known.
Response: Thanks very much for your question. The corresponding contents have been added in the manuscript. A hybrid paper mulberry (Broussonetia kazinoki × Broussonetia papyrifera), which belongs to Moraceae family, has a great adaptability to climates and soils, rapid growth rate, multi-resistance to pests and diseases, and high protein content in leaves.
- Keywords should not be the same as words already used in a title.
Response: Thank you very much for your comments. We agree with your point and have changed the keywords.
- “Differentially accumulated proteins”, “Protein differences”, “Differential accumulated proteins”, and “DAPs” are used exchange ably. It should be unified and corrected.
Response: Thanks very much for your question. We have modified the article according to your suggestion.
- Figure 8 should be smaller.
Response: Thanks very much for your kind reminding. Figure 8 was replaced according to your suggestion, please check it.
- Line 313-314. The sentence “These findings indicated that the molecular mechanism of the golden-yellow mutant was complicated” does not sound scientifically good, and it looks like a response of a student to the tricky question because, after all, everything is complicated. (Please excuse this joke.)
Response: We apologize for this mistake and we correct this sentence. “Therefore, the formation mechanism of golden-yellow mutants requires further investigation.” Please check lines 346-347.
- Line 424. The sentence is incorrect because ALA content cannot be extracted. Extracted can be ALA, and its content may be determined.
Response: We apologize for this mistake and we correct this sentence. “The 5-aminolevulinic acid (ALA) content was determined as described by Dei”. Please check lines 469-470.

Round 2
Reviewer 1 Report
Authors improved the manuscript.